# Auxiliary subunits of the CKAMP family differentially modulate AMPA receptor properties

Paul Farrow[1,2†], Konstantin Khodosevich[3,4†§], Yechiam Sapir[5†],
Anton Schulmann[3,4†], Muhammad Aslam[1,2], Yael Stern-Bach[5‡*],
Hannah Monyer[3‡*], Jakob von Engelhardt[1,2‡*]

[1]Synaptic Signalling and Neurodegeneration, German Cancer Research Center, Heidelberg, Germany; [2]Synaptic Signalling and Neurodegeneration, German Center for Neurodegenerative Diseases, Bonn, Germany; [3]Department of Clinical Neurobiology, Medical Faculty of Heidelberg University, Heidelberg, Germany; [4]German Cancer Research Center, Heidelberg, Germany; [5]Department of Biochemistry and Molecular Biology, Institute for Medical Research – Israel-Canada, The Hebrew University-Hadassah Medical School, Jerusalem, Israel

**\*For correspondence:**
yaelste@ekmd.huji.ac.il (YS-B);
h.monyer@dkfz-heidelberg.de (HM);
engelhardt@dzne.de (JE)

[†]These authors contributed equally to this work
[‡]These authors also contributed equally to this work

**Present address:** [§]BRIC – Biotech Research and Innovation Center, Copenhagen University, Copenhagen, Denmark

**Competing interests:** The author declares that no competing interests exist.

**Abstract** AMPA receptor (AMPAR) function is modulated by auxiliary subunits. Here, we report on three AMPAR interacting proteins—namely CKAMP39, CKAMP52 and CKAMP59—that, together with the previously characterized CKAMP44, constitute a novel family of auxiliary subunits distinct from other families of AMPAR interacting proteins. The new members of the CKAMP family display distinct regional and developmental expression profiles in the mouse brain. Notably, despite their structural similarities they exert diverse modulation on AMPAR gating by influencing deactivation, desensitization and recovery from desensitization, as well as glutamate and cyclothiazide potency to AMPARs. This study indicates that AMPAR function is very precisely controlled by the cell-type specific expression of the CKAMP family members.
DOI: https://doi.org/10.7554/eLife.09693.001

## Introduction

AMPARs mediate the majority of fast excitatory transmission in the central nervous system and play a key role in brain plasticity. AMPAR function is controlled by a multitude of auxiliary subunits (*Yan and Tomita, 2012*). These include TARPs (*Tomita et al., 2003*), cornichons (*Schwenk et al., 2009*), Sol-1 (*Zheng et al., 2004*) and SynDIG1 (*Kalashnikova et al., 2010*). Recently, we identified a novel AMPAR auxiliary subunit, CKAMP44, and characterized its modulation of AMPAR gating properties in CA1 and dentate gyrus neurons (*Khodosevich et al., 2014*; *von Engelhardt et al., 2010*). Unlike other auxiliary subunits, CKAMP44 contains an N-terminal cystine-knot domain that in other proteins, e.g. growth factors (*McDonald and Hendrickson, 1993*), was shown to stabilize the globular structure of the protein. The different auxiliary subunits exhibit distinct modulatory profiles. Since auxiliary subunits are differentially expressed in the brain, the specific combination in a particular cell type is likely to govern the AMPAR response to glutamate, as is the case for dentate gyrus granule cells, which express TARP γ-8 and CKAMP44. Both proteins increase the number of AMPARs on the cell surface, decrease the deactivation rate and increase glutamate affinity. However, they differ in the influence that they extend on AMPAR desensitization, recovery from desensitization, long-term and short-term potentiation (*Khodosevich et al., 2014*).

Here, based on homology with CKAMP44, we report on three novel CKAMP44-like proteins that were named CKAMP39, CKAMP52 and CKAMP59 and, together with CKAMP44, constitute the

**eLife digest** The brain processes and transmits information through large networks of cells called neurons. A neuron can pass the information it receives to other neurons by releasing chemicals called neurotransmitters across junctions known as synapses. These chemicals bind to receptor proteins on the surface of the neighboring neuron, which triggers changes that affect the activity of this neuron.

Glutamate is the most commonly used neurotransmitter in the brain and binds to receptor proteins called AMPA receptors. If a neuron frequently sends glutamate across a particular synapse, the number of AMPA receptors in the second neuron will increase in response. This makes signaling across the synapse easier – a process known as synaptic strengthening. The ability to change the strength of synapses is important for learning and memory.

Proteins called auxiliary subunits also bind to AMPA receptors and regulate their properties, and hence also affect the strength of the synapse. For instance, some auxiliary subunits increase the number of AMPA receptors at the synapse, while others have an effect on how the receptor protein works. In 2010, researchers identified a new auxiliary protein called CKAMP44 that modifies AMPA receptor activity. Now, Farrow, Khodosevich, Sapir, Schulmann et al. – including some of the researchers involved in the 2010 study – have identified three other auxiliary proteins that are similar to CKAMP44. Collectively, these four proteins are termed the CKAMP family.

The sequences of all four proteins were found to share many common features, especially in the regions that bind to the AMPA receptors. Like CKAMP44, the new members of the CKAMP family are only present in the brain, although each protein is produced in different brain regions. Further investigation revealed that each member of the CKAMP family affects the AMPA receptor channels in a different way.

Taken together, Farrow et al.'s results suggest that the different CKAMP family members allow the activity of the AMPA receptors to be precisely controlled. The next challenge is to understand in more detail how each CKAMP family member influences how AMPA receptors work.
DOI: https://doi.org/10.7554/eLife.09693.002

CKAMP family. Like CKAMP44, the newly identified CKAMPs are all single transmembrane domain proteins that possess an extracellular cystine-knot domain and an intracellular domain ending with a PDZ type II motif. Notably, novel CKAMP family members bind to GluA1 and GluA2 and modify AMPAR-mediated currents in heterologous expression systems.

## Results

### Identification of novel *CKAMP* genes in the mouse genome

To investigate whether CKAMP44 has homologues in rodents, we searched the genomic databases using either the complete sequence of the *CKAMP44* gene or the *CKAMP44* cystine-knot domain sequence as a reference. We found three genes with a high degree of similarity to CKAMP44 and named them according to the predicted molecular weight of their corresponding protein products—CKAMP39, CKAMP52 and CKAMP59 (*Figure 1A*). Due to the similarity in their peptide sequences (especially the cystine-knot motif), we classified these four proteins as the CKAMP family members.

Like CKAMP44, all CKAMP proteins comprise a signal peptide, N-terminal cystine-knot extracellular domain, transmembrane domain and a large intracellular C-terminal domain, which terminates in a PDZ type II motif (*Figure 1A,B*). The predicted signal peptides vary in length, being 36, 30 and 22 amino acids for CKAMP39, CKAMP52 and CKAMP59, respectively (*Figure 1B*). Strikingly, cystine-knot domains exhibit a high degree of similarity and all eight cysteines are conserved among the family members (*Figure 1B*). Although cystine-knot containing proteins differ much in their function (*Heinemann and Leipold, 2007*; *McDonald and Hendrickson, 1993*; *Zhang et al., 2011*), the purpose of cysteine-knots is similar, i.e. to form a tightly packed globular domain made of β-strands with several variable loops for stable protein-protein interaction. Thus, although sequences of the extracellular domains of CKAMP proteins have little homology beyond the cystine-knot core, it is likely that the extracellular region of all four proteins exhibit a similar β-strand structure. Each of the

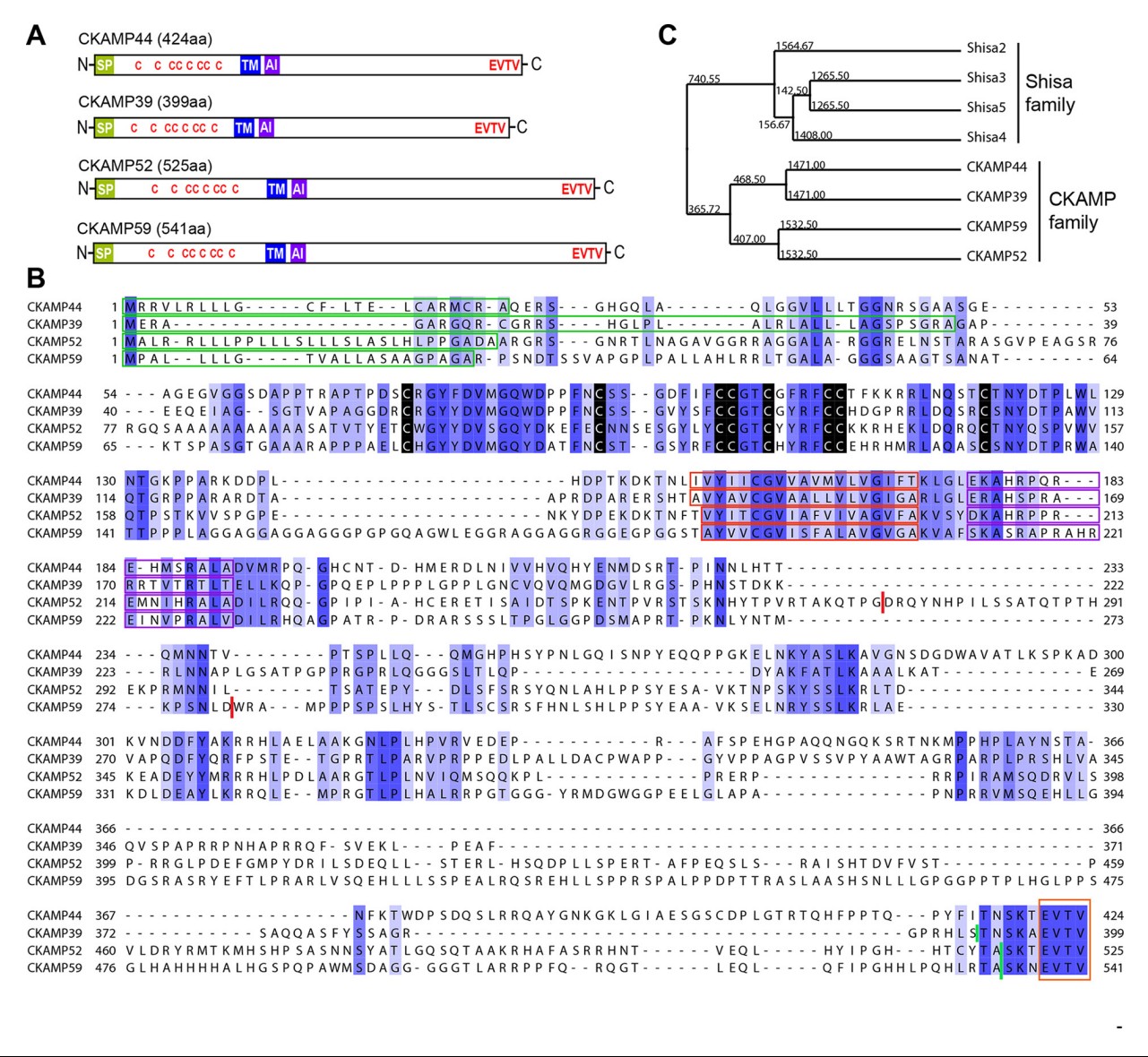

**Figure 1.** Identification and molecular characterization of CKAMP proteins. (**A**) Schematic drawing of CKAMP proteins, depicting the signal peptide (SP), cysteines (C) of the cystine knot, transmembrane domain (TM) and PDZ type II motif (EVTV). 'AI' indicates "AMPAR interacting" region, based on the GluA1 binding co-IP experiments from *Khodosevich et al. (2014)*. (**B**) Protein sequence alignment of mouse CKAMP39, CKAMP44, CKAMP52 and CKAMP59. Amino acids marked in blue are similar or identical among CKAMP family members. Intensity of the blue color indicates the degree of similarity, with identical amino acid positions in the protein sequence highlighted by the most intense color. Green rectangles outline the predicted signal peptides, red rectangles—the predicted transmembrane domains, purple rectangles - putative AMPAR interacting regions and orange rectangle—the PDZ type II motif. Positions of cysteines belonging to the cystine-knot motif are indicated in black. Thick red lines in position 274 for CKAMP52 and 280 for CKAMP59 indicate insertion sites for additional aa sequences that are encoded by alternatively spliced exon 3 and exon 4, respectively. Thick green lines nearby C-termini indicate positions for flag-tag insertions. (**C**) Phylogenetic analysis of Shisa and CKAMP proteins, based on their protein sequence (average distance tree).

DOI: https://doi.org/10.7554/eLife.09693.003

The following figure supplement is available for figure 1:

**Figure supplement 1.** Comparison of Shisa proteins and CKAMPs.

DOI: https://doi.org/10.7554/eLife.09693.004

proteins possesses a predicted single short transmembrane domain (18–19 amino acids), and novel CKAMP family members have ~80% homology with the transmembrane domain of CKAMP44 (*Figure 1B*). Notably, a ~20 amino acid stretch immediately downstream of the transmembrane

domain contains a conserved arginine-rich motif (*Figure 1B*). This is of particular interest, since this region of CKAMP44 was shown to be necessary for interaction with GluA1, and CKAMP44 mutants with a deletion of only 6 amino acids in this region did not bind to GluA1 when overexpressed in HEK293/T17 cells (*Khodosevich et al., 2014*). Finally, another conserved region in the protein sequence amongst CKAMP family members is the PDZ type II motif located at the very end of the C-terminal domain (*Figure 1B*). Thus, all CKAMP proteins end with the same EVTV stretch. Furthermore, another 5 amino acids upstream of the PDZ motif are also almost identical in CKAMP proteins. The PDZ motif is of functional importance as it mediates the interaction of CKAMP44 with PSD95, which allows anchoring of AMPARs within synapses of dentate gyrus granule cells (*Khodosevich et al., 2014*).

At the time of CKAMP family identification, the genes of the CKAMP family members had not yet been recorded in the Refseq database. Currently, based on their similarity to *Shisa* genes, the *Ckamp* genes are termed in the genome database as *Shisa6* (CKAMP52), *Shisa7* (CKAMP59), *Shisa8* (CKAMP39) and *Shisa9* (CKAMP44). Shisa 1–3 proteins were studied in embryonic development of *Xenopus laevis*, where they were found to inhibit Wnt and FGF signaling by retention of their receptors in ER (*Nagano et al., 2006*; *Yamamoto et al., 2005*). However, CKAMP family members differ significantly from Shisa2-5 proteins (*Figure 1—figure supplement 1*, Shisa1 protein does not have a mouse homolog) and form a separate cluster on the phylogenetic tree (*Figure 1C*, see also *Pei and Grishin, 2012*). There are ~70 amino acids that are largely conserved in the CKAMP cluster, but not in the Shisa cluster (*Figure 1—figure supplement 1*). Furthermore, Shisa proteins are much shorter than CKAMPs, being 197-295 and 399-541 amino acids, respectively. Interestingly, the GluA1-interacting region of CKAMP44 exhibits a high degree of similarity to other CKAMPs, but not to that of Shisa proteins (*Figure 1—figure supplement 1*). Finally, Shisa proteins do not contain a PDZ binding motif at their C-terminus. Based on these considerations, we propose that CKAMP members constitute a protein family distinct from the Shisa protein family (*Figure 1C*).

## Novel CKAMP proteins are expressed in the mouse brain and interact with AMPARs in HEK293/T17 cells

We amplified open reading frames (ORFs) for the three novel CKAMP proteins using mouse brain-derived mRNA and confirmed the sequence of the corresponding proteins. Since at least 30 clones per CKAMP family member were analyzed, we were able to estimate the relative expression levels of different CKAMP splice isoforms in the brain. While CKAMP39 had only one splice variant, both CKAMP52 and CKAMP59 had two splice isoforms (*Figure 2—figure supplement 1*) that differed in protein coding sequences. In the subsequent experiments, we utilized the most abundant versions of CKAMPs, i.e. CKAMP52 and CKAMP59 lacking exon 3 and exon 4, respectively (note that alignments in *Figure 1B* and *Figure 1—figure supplement 1* are performed for exon 3- and exon 4-lacking versions). Exon 3 of CKAMP52 and exon 4 of CKAMP59 encode 32 and 17 intracellular amino acids, respectively, downstream of the AMPAR-interacting domain and upstream of the PDZ motif. Interestingly, there are also two splice variants of CKAMP44 (*von Engelhardt et al., 2010*), but alternatively spliced mRNA was not reported for mouse Shisa family members.

Previously, we demonstrated that CKAMP44 is expressed exclusively in the brain (*von Engelhardt et al., 2010*). Based on gene expression database BioGPS, we found that the new members of the CKAMP family are also expressed exclusively in the brain (*Figure 2—figure supplement 2*). In situ hybridization of adult mouse brain sections with oligo probes against CKAMP39, CKAMP52 and CKAMP59 mRNAs revealed that each of the novel CKAMP family proteins exhibited a region-specific expression pattern within the brain (*Figure 2A*). CKAMP39 expression was restricted to two brain regions, namely the cerebellum and olfactory bulb, which were also the only brain regions with significant CKAMP39 expression according to the BioGPS database (*Figure 2—figure supplement 2*). Both CKAMP52 and CKAMP59 were expressed in the hippocampus, but CKAMP59 was also expressed in the cortex and olfactory bulb, whereas CKAMP52 was expressed in the cerebellum and septum. CKAMP39 is absent and CKAMP52 is barely detectable in the brain of embryonic day 17 (E17) mice. In contrast, there is a strong signal for CKAMP59 already prenatally. Postnatally, there is little change in the expression pattern of any of these CKAMPs, except for an upregulation of CKAMP39 and CKAMP52 in the cerebellum and olfactory bulb and a modest downregulation of CKAMP59 in the thalamus and brainstem (*Figure 2A*).

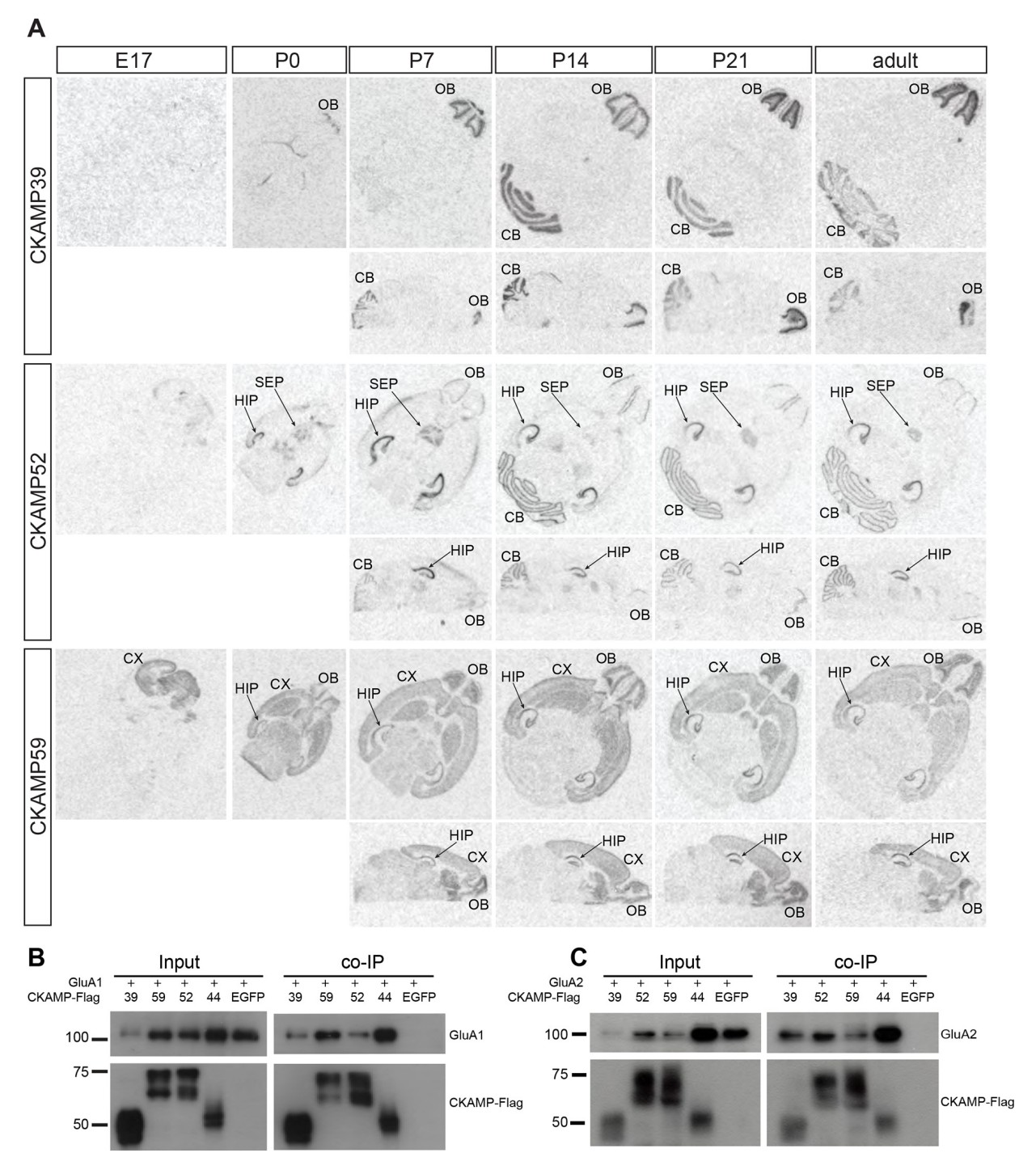

**Figure 2.** Expression pattern of CKAMPs and their interaction with AMPARs. (A) Transcription of *CKAMP* genes as visualized by in situ hybridization of brain sections obtained from mice at different developmental ages. Abbreviations: CB - cerebellum, CX - cortex, HIP - hippocampus, OB - olfactory bulb, SEP - septum. (B) and (C) All members of the CKAMP family interact with GluA1 and GluA2, respectively. HEK293/T17 cells were transfected with GluA1 (B) or GluA2 (C) together with one of the indicated flag-tagged CKAMPs or with EGFP as a control. Proteins were immunoprecipitated using anti-flag antibody. All flag-tagged CKAMPs, but not EGFP, co-precipitated GluA1 (B) or GluA2 (C) from the total protein fraction. Input corresponds to ~9% of GluA1 or ~3% of GluA2 co-immunoprecipitation experiment.

DOI: https://doi.org/10.7554/eLife.09693.005

The following figure supplements are available for figure 2:

*Figure 2 continued on next page*

*Figure 2 continued*

**Figure supplement 1.** Analysis of splice isoforms for CKAMP family members.
DOI: https://doi.org/10.7554/eLife.09693.006
**Figure supplement 2.** Analysis of expression for CKAMP family members.
DOI: https://doi.org/10.7554/eLife.09693.007

To determine whether CKAMP39, CKAMP52 and CKAMP59 interact with AMPARs, we inserted a flag-tag at the non-conserved C-terminal part of the proteins, and co-expressed flag-tagged CKAMPs along with GluA1 or GluA2 in HEK293/T17 cells. With an anti-flag antibody, GluA1 and GluA2 co-immunoprecipitated from protein samples of HEK293/T17 cells co-expressing flag-tagged CKAMP39, CKAMP52 or CKAMP59 (*Figure 2B*), showing that all novel CKAMPs bind to GluA1 and GluA2 in a heterologous expression system. All CKAMPs had two bands on Western blot, indicating their likely glycosylation that was shown previously for CKAMP44 (*von Engelhardt et al., 2010*).

## Novel CKAMP members modify AMPAR-mediated currents in heterologous expression systems

To characterize the functional consequences of the interaction between the CKAMP family members and AMPARs, we performed electrophysiological experiments employing *Xenopus Laevis* oocytes. To investigate how gating properties are modulated by the novel CKAMPs, we performed fast perfusion patch-clamp recordings on outside-out macropatches pulled from oocytes expressing either GluA1 or GluA2(Q) alone, or with CKAMP39 or CKAMP52. An analysis of the effect of CKAMP59 on AMPAR gating in oocytes was not possible, as this auxiliary subunit was not sufficiently expressed in this heterologous expression system as revealed by the absence of detectable protein in Western-blot analysis (*Figure 3A*). Consistently, no significant change in AMPAR gating was observed for the co-expression of GluA1 or GluA2(Q) with CKAMP59. In contrast, CKAMP39 and CKAMP52 exhibited protein levels comparable to CKAMP44 (*Figure 3A*), and influenced AMPAR gating properties differentially. Neither CKAMP39 nor CKAMP52 modulated the GluA1-mediated deactivation time constant ($\tau_{deact}$), but both increased $\tau_{deact}$ of GluA2(Q)-mediated currents (*Figure 3B* and *Supplementary file 1A—table 1*). Both proteins also had no influence on the GluA1-mediated desensitization time constant ($\tau_{des}$), but significantly reduced $\tau_{des}$ of GluA2(Q)-mediated currents (*Figure 3C*). There was a trend towards increased steady-state current amplitude (as a percentage of maximal current) during the 500 ms glutamate application in the oocyte patches for GluA1 with CKAMP52 (see below for the significant effect on steady-state currents in HEK293/T17 cells), and a significant reduction in steady-state current amplitude of GluA2(Q)-mediated currents by CKAMP39 (*Figure 3C* and *Supplementary file 1A—table 1*). The time constant of recovery from desensitization ($\tau_{recovery}$) of GluA1- and GluA2(Q)-mediated currents was increased by CKAMP39, whereas there was only a small increase and decrease of $\tau_{recovery}$ of GluA1- and GluA2(Q)-mediated currents, respectively, when co-expressing CKAMP52 (*Figure 3D* and *Supplementary file 1A—table 1*).

We previously showed that CKAMP44 increases glutamate potency (*von Engelhardt et al., 2010*). A comparable decrease in glutamate EC50 was observed when GluA1 or GluA2 was expressed with CKAMP39 or CKAMP52. The most dramatic change was seen when expressing GluA2 together with CKAMP52; the EC50 was more than 10 fold smaller for CKAMP52-bound GluA2 compared to GluA2 alone (*Figure 4A*). CKAMP39 and CKAMP44, but not CKAMP52, influenced not only the potency of glutamate, but also that of the AMPAR desensitization blocker cyclothiazide (CTZ). Thus, there was an increase in the CTZ EC50 when co-expressing GluA1 with CKAMP39, and GluA2 with CKAMP39 or CKAMP44 (*Figure 4B* and *Supplementary file 1B—table 2*).

CKAMP59 is well expressed in HEK293/T17 cells as revealed by Western blot analysis (*Figure 2B*) in contrast to the oocytes. Hence, to investigate the influence of CKAMP59 on AMPAR-mediated currents, we expressed this auxiliary subunit along with GluA1 or GluA2(Q) in HEK293/T17 cells. To be able to compare the influence of CKAMP59 with that of the other auxiliary subunits, we also investigated AMPAR-mediated currents in HEK293/T17 cells that co-expressed CKAMP39 or CKAMP52. In contrast to the other auxiliary subunits, CKAMP59 did not modulate GluA1- or GluA2(Q)-mediated current kinetics. However, there was a significant reduction in GluA2(Q)-mediated current amplitude. Co-expression of CKAMP39 and CKAMP52 also reduced AMPAR-mediated current

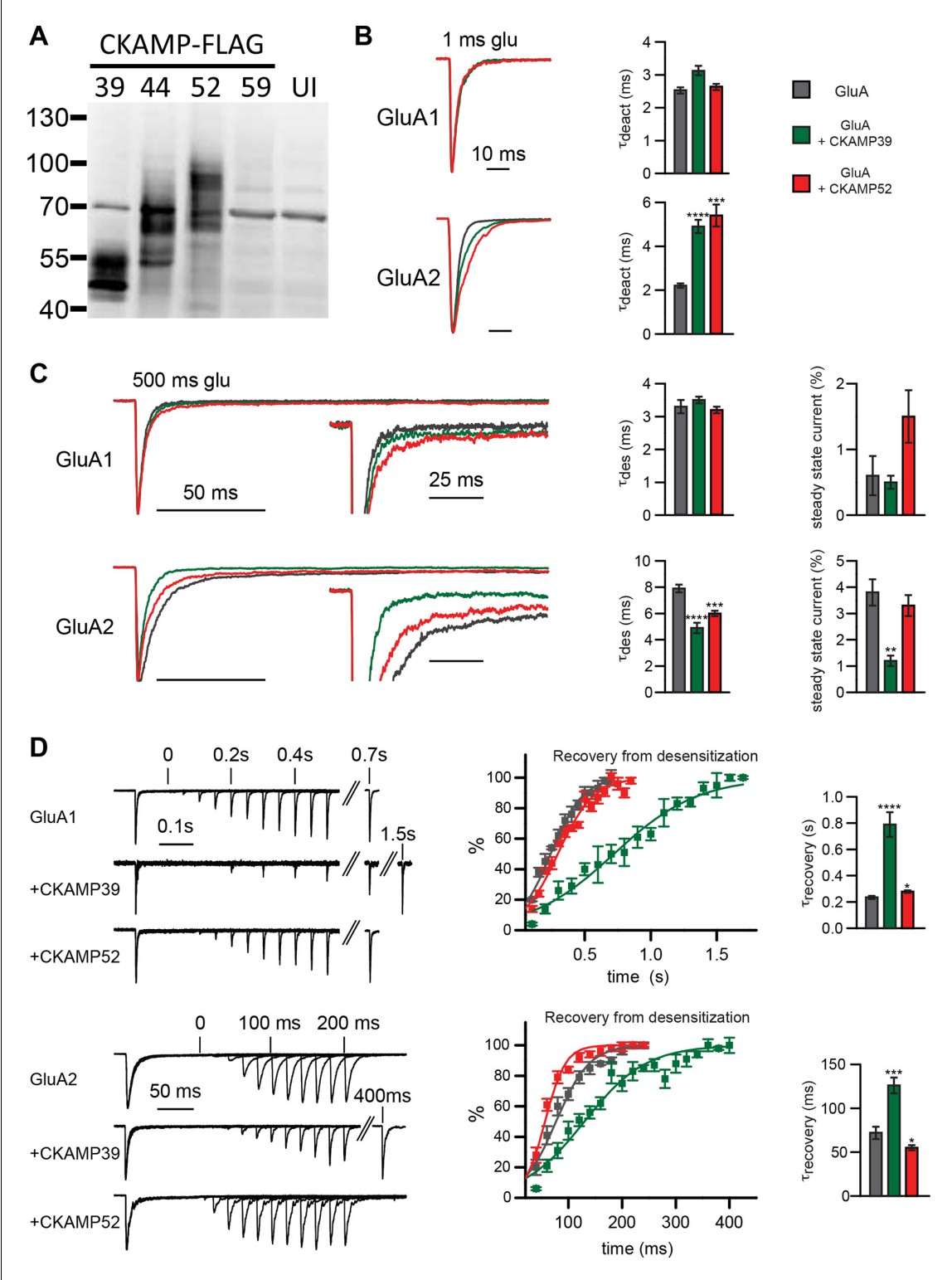

**Figure 3.** Modulation of AMPAR-mediated currents in Xenopus laevis oocytes. (**A**) Western blot analysis on flag-tagged CKAMPs in Xenopus laevis oocytes. CKAMP39, CKAMP44, CKAMP52 & CKAMP59 where injected at 1, 3, 5 & 10 ng/oocyte, respectively (UI-un-injected oocytes). (**B**) Deactivation rate ($\tau_{deact}$), (**C**) desensitization rate ($\tau_{des}$) and steady-state amplitude and (**D**) weighted time constant of recovery from desensitization ($\tau_{recovery}$) of GluA1- and GluA2(Q)-mediated currents. Deactivation and desensitization were tested by application of 10 mM glutamate for 1 ms and 500 ms, respectively, onto macropatches of oocytes. $\tau_{recovery}$ was estimated with application of two 100 ms glutamate pulses with different inter-pulse intervals. Example currents are shown on the left of the quantification (mean ± SEM).

DOI: https://doi.org/10.7554/eLife.09693.008

amplitude (GluA1-current amplitude was reduced only by CKAMP39) (*Figure 5A* and *Supplementary file 1C—table 3*). A cell surface biotinylation assay performed in HEK293/T17 cells co-transfected with AMPARs and CKAMPs showed that all CKAMP family members, except for CKAMP52, lead to a reduction in surface expression of GluA1 and GluA2 protein. These results can be accounted for by a reduction of total expression of GluA1 and GluA2, and by reduced GluA2 forward trafficking or stabilization of GluA2 on the cell surface as indicated by the reduced ratio of surface to total protein. Interestingly, CKAMP52 increased the ratio of surface to total GluA1 expression, suggesting that this auxiliary protein exerts an opposite influence on forward trafficking or stabilization of GluA1 and GluA2 (*Figure 5—figure supplement 1* and *Supplementary file 1D—table 4*).

The strong reduction of surface AMPAR expression precluded an analysis of current kinetics by fast-application of glutamate onto outside-out patches. Thus, we performed an analysis using fast-application of glutamate onto whole HEK293/T17 cells instead. The expected solution exchange time is considerably slower when using whole cells instead of outside-out patches (*Barberis et al., 2008*). Nevertheless, the analysis allowed us to draw conclusions about the influence of CKAMP family members on AMPAR kinetic properties. Thus, CKAMP39 and CKAMP52 modulated AMPAR gating in HEK293/T17 cells similarly to what we observed in oocytes with an increase in GluA2(Q) $\tau_{deact}$ by CKAMP52 (*Figure 5B* and *Supplementary file 1C—table 3*), a decrease in GluA2(Q) $\tau_{des}$ by CKAMP39, an increase in GluA1 steady-state current amplitude by CKAMP52, and a reduction of the GluA2(Q) steady-state current amplitude by CKAMP39 (*Figure 5C* and *Supplementary file 1C—table 3*). Recovery from desensitization was analyzed in HEK293/T17 cells with a protocol that differed from that used in oocyte experiments, where we applied two 100 ms glutamate pulses with different interpulse intervals. In HEK293/T17 cell experiments, we applied two 1ms glutamate pulses. The rational was to probe whether CKAMP39 influences AMPAR recovery from desensitization also when glutamate is applied only for very short time periods, thus mimicking the short presence of glutamate in the synaptic cleft. Indeed, CKAMP39 slowed the recovery from desensitization of GluA1- and GluA2(Q)-mediated currents also when tested with this modified protocol (*Figure 5D* and *Supplementary file 1C—table 3*). The effect was comparable to that of CKAMP44, which modulates synaptic short-term plasticity in dentate gyrus granule cell synapses by slowing recovery from desensitization (*Khodosevich et al., 2014*).

There were some differences in the modulation of GluA2(Q)-mediated currents in HEK293/T17 cells and oocytes. Thus, CKAMP39 increases $\tau_{deact}$ in oocytes, but not in HEK293/T17 cells. In addition, $\tau_{des}$ was reduced by co-expression with CKAMP52 in oocytes, but not in HEK293/T17 cells. Finally, the steady-state current amplitude was increased by CKAMP52 in HEK293/T17 cells, but not in oocytes. In conclusion, the new CKAMP family members display very distinct modulatory effects on AMPAR gating. Like the prototypical TARP auxiliary subunits (*Kato et al., 2008*), the CKAMP family affects AMPAR in a subunit-specific manner.

## Discussion

In recent years, several labs have employed large proteomic screens to search for AMPAR interacting proteins, which resulted in the identification of new AMPAR auxiliary (or auxiliary-like) subunits, such as CKAMP44 (*von Engelhardt et al., 2010*), cornichons (*Schwenk et al., 2009*) and GSG1L (*Schwenk et al., 2012*; *Shanks et al., 2012*). In this study, we searched a genomic and transcriptomic databases, and identified three new proteins that, together with CKAMP44, form the CKAMP family of AMPAR auxiliary-like proteins. Presumably, evolutionarily the CKAMPs and Shisa proteins descend from the same protein family. The Shisa proteins were shown to be involved in fibroblast growth factor receptor maturation and degradation during embryogenesis in *Xenopus laevis* oocytes (*Nagano et al., 2006*; *Yamamoto et al., 2005*). All CKAMPs exhibited a significant homology in the region of CKAMP44 that is necessary for AMPAR binding (*Khodosevich et al., 2014*). Thus, it is likely that, similar to CKAMP44, a stretch of amino acids immediately downstream of the transmembrane regions of novel CKAMP proteins is involved in interaction with AMPARs. All CKAMPs possess an extracellular cystine-knot motif that was shown to be important for modulation of AMPAR gating, and an identical intracellular PDZ-domain binding motif (*Khodosevich et al., 2014*). All CKAMPs bind to GluA1 and GluA2 and modify GluA1 and/or GluA2 currents in two heterologous systems, making them likely candidates for AMPAR auxiliary subunits in vivo. Importantly, despite their

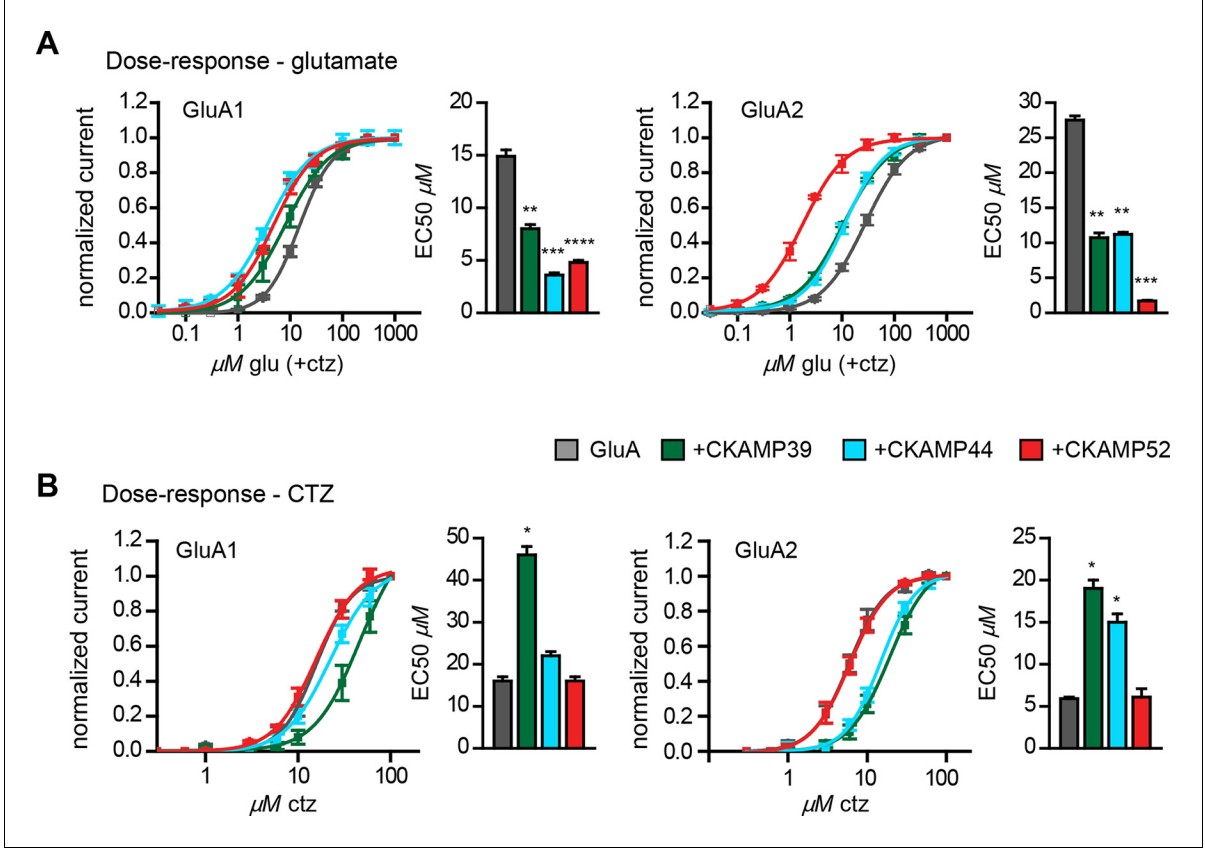

**Figure 4.** Influence of CKAMP proteins on glutamate and CTZ potency. (**A**) Glutamate potency for GluA1 or GluA2(Q) expressed along with CKAMP39, CKAMP44 or CKAMP52 in Xenopus laevis oocytes. Responses to increasing concentration of glutamate were recorded in the presence of 0.1 mM CTZ applied 10 s before and during glutamate application to reach equilibrium steady-state currents. Responses were normalized to the response obtained for 1mM glutamate. (**B**) CTZ potency for GluA1 or GluA2(Q) expressed along with CKAMP39, CKAMP44 or CKAMP52 in Xenopus laevis oocytes. Responses to 1 mM glutamate were recorded in the presence of increasing concentrations of CTZ after a 10 s incubation of the oocyte with the respective CTZ concentration without glutamate. Current responses were normalized to the response obtained with 0.1 mM CTZ after subtraction of the current induced by 1 mM glutamate without CTZ. The respective EC50 values (mean ± SEM) were calculated from fits to data obtained from individual oocytes.

DOI: https://doi.org/10.7554/eLife.09693.009

structural similarities, the members of the CKAMP family differ enormously in their modulation of AMPAR-mediated currents. Only the function of CKAMP39 resembles that of CKAMP44 (*Khodosevich et al., 2014*), and indeed this subunit also displays the highest homology with CKAMP44. CKAMP39 had a pronounced effect on the recovery from desensitization, similar to CKAMP44, suggesting that it might also modulate synaptic short-term plasticity as was observed for CKAMP44 in dentate gyrus granule cells (*Khodosevich et al., 2014*). The reduction of surface AMPAR number by all three CKAMPs in HEK293/T17 cells was unexpected, since CKAMP44 has the opposite effect on AMPARs of granule cells, possibly by promoting trafficking of AMPARs to the cell surface. The reduction in surface AMPAR number by the novel CKAMP family members was mainly due to a reduction in total AMPAR number in the cell. However, changes in the ratio of surface/total AMPAR number indicate that the novel CKAMP family members also influence the forward trafficking or stability of AMPARs on the cell surface.

The fact that some parameters of GluA2(Q)-mediated currents were modulated only in HEK293/T17 cells, others only in oocytes, cannot be explained by differences in AMPAR-composition, since the same GluA2(Q)-flip version of this subunit was expressed in both expression systems. However, one explanation could be the different recording conditions used for HEK293/T17 cells and oocytes. Thus, there were differences in room temperature (17°C versus 22°C), patch size (macropatches of oocytes versus lifted whole HEK293/T17 cells) and the holding potential (-70 versus -120 mV).

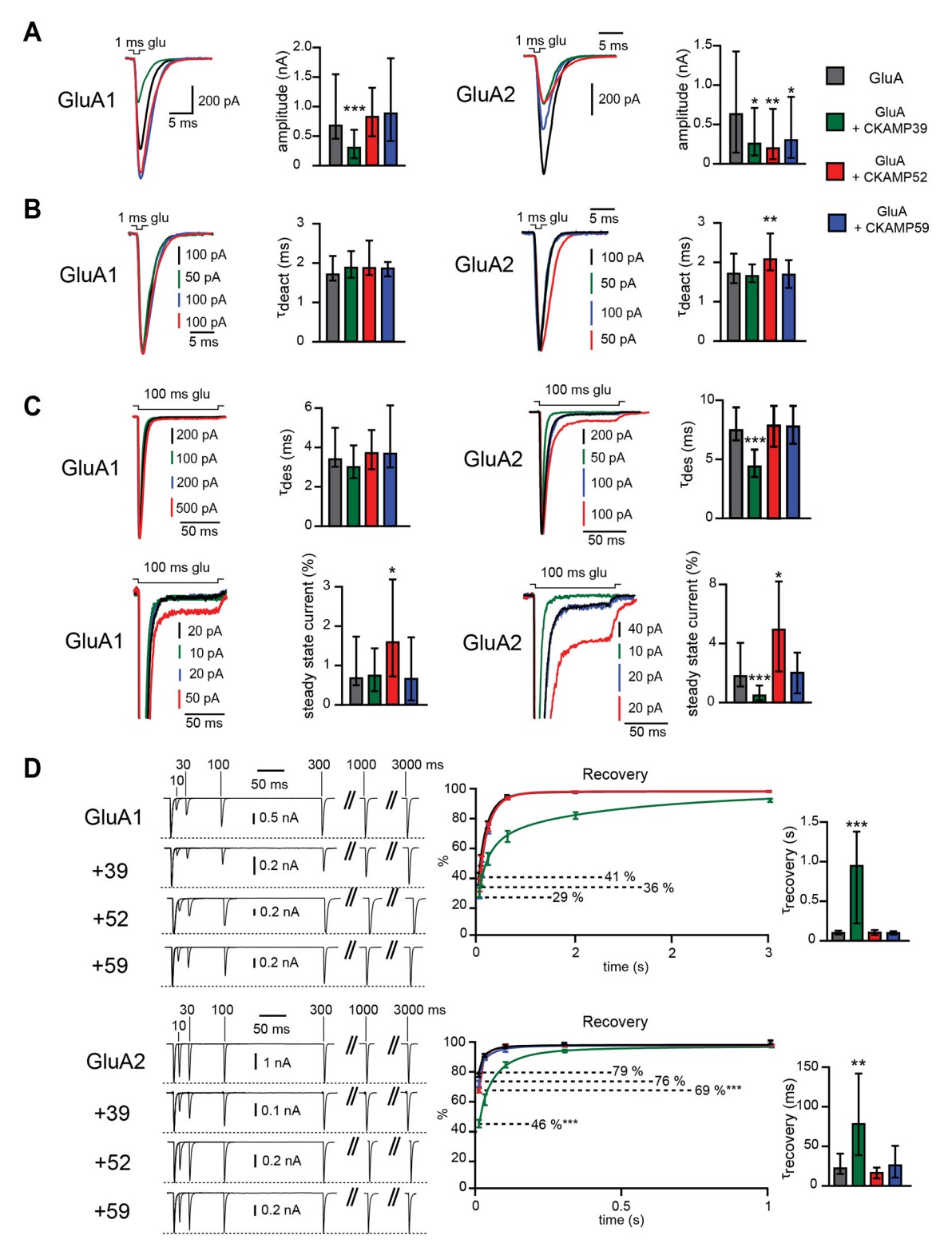

**Figure 5.** Modulation of AMPAR-mediated currents in HEK293/T17 cells. (**A**) Peak current amplitude, (**B**) deactivation rate ($\tau_{deact}$), (**C**) desensitization rate ($\tau_{des}$) and steady-state amplitude and (**D**) $\tau_{recovery}$ of GluA1- and GluA2(Q)-mediated currents. Deactivation and desensitization were tested by application of 1 mM glutamate for 1 ms and 100 ms, respectively. $\tau_{recovery}$ was analyzed with application of two 1 ms glutamate pulses with different inter-pulse intervals. Example currents are shown on the left of the quantification (median ± IQR).

DOI: https://doi.org/10.7554/eLife.09693.010

*Figure 5 continued*

The following figure supplement is available for figure 5:

**Figure supplement 1.** Analysis of total and surface AMPAR expression in HEK293/T17 cells.
DOI: https://doi.org/10.7554/eLife.09693.011

Overall, the novel CKAMP family members greatly expand the pool of AMPAR auxiliary-like proteins expressed in the brain. CKAMP52 was also identified as an AMPAR interacting protein in recent proteomic screens (*Schwenk et al., 2012*; *Shanks et al., 2012*), showing that at least this CKAMP family member interacts with AMPARs in the brain. It is not clear why CKAMP39 and CKAMP59 were not identified as AMPAR interacting proteins, but it is possible that their expression level is too low or that their interaction is too loose for identification in the proteomic screens. The highly diverse expression pattern of CKAMP family members together with their unique biophysical profile make them strong candidates for region-specific AMPAR modulation. Future studies should unravel how different CKAMPs influence synaptic function by modulating expression and gating kinetics of AMPARs. Yet another effect of CKAMP family members might be an influence on neuron morphology as described for CKAMP44 and TARP γ-8, which increase spine number by augmenting surface AMPAR expression (*Khodosevich et al., 2014*).

## Materials and methods

### Bioinformatics analysis

Signal peptides were analyzed using SignalP 4.1 software (*Petersen et al., 2011*) (http://www.cbs.dtu.dk/services/SignalP/). Transmembrane domains were identified using TMHMM 2.0 software (*Krogh et al., 2001*) (http://www.cbs.dtu.dk/services/TMHMM/) and PredictProtein (*Yachdav et al., 2014*) (www.predictprotein.org). Alignment of protein sequences and phylogenetic analysis was performed using Jalview (*Troshin et al., 2011*; *Waterhouse et al., 2009*) (www.jalview.org). The alignments of CKAMP proteins and of CKAMP and Shisa proteins were done using ProbCons (*Do et al., 2005*) and Clustal (*Larkin et al., 2007*) tools, respectively. The phylogenetic tree of Shisa proteins and CKAMPs was obtained utilizing average distance methods. Expression of CKAMP family members in different tissues was retrieved from gene expression BioGPS database (http://biogps.org/), GNF1M dataset for CKAMP39 and MOE430 dataset for CKAMP52 and CKAMP59.

### Molecular cloning

RNA was isolated from the whole mouse brain and cDNA was synthesized as described before (*Khodosevich et al., 2013*). We used the following primers to amplify CKAMP39, CKAMP52 and CKAMP59:

 C39F = 5'TAGGATCCGCCACCATGGAGCGCGCTGGGGCGCGGGGACAG
 C39R = 5'GCACTAGTCTAGACCGTGACCTCGGCTTTGC
 C52F = 5'ATGGATCCGCCACCATGGCGCTGCGCCGCCTCCTG
 C52R = 5'CGACTAGTTCACACGGTCACTTCAGTCTTGCTGGC
 C59F = 5'TAGGATCCGCCACCATGCCGGCCCTGCTGCTGCTC
 C59R = 5'GCCACTAGTTCAGACAGTCACTTCGTTCTTGCTG

CKAMPs were amplified from cDNA using LA Taq polymerase with GC-rich buffer (Clontech-Takara Bio, France) for 5 cycles. PCR products were purified and re-amplified using PfuUltraII (Agilent, USA) for 30 more cycles. The resulting PCR products were digested by BamHI and SpeI and ligated into the pRK5 vector or an AAV vector with the human synapsin promoter, IRES and EGFP (pAAV-Syn-IRES-EGFP) (*von Engelhardt et al., 2010*). CKAMP ORFs were sequenced and only those that corresponded to sequences from the genomic database were used in the subsequent experiments. To estimate the abundance of different splice-variants in the brain, at least 30 clones per each CKAMP family member were analyzed.

To generate flag-tagged versions of CKAMPs, we inserted a flag-tag sequence close to the non-conserved terminal region of CKAMPs (upstream to PDZ type II motif). The insertions were made using site-directed mutagenesis via two consecutive rounds of extension PCR, re-ligating flag-

tagged PCR fragments using BspEI/SalI for CKAMP39 and XhoI/SalI for CKAMP52 and CKAMP59. The primers that we utilized for flag-tag insertion are shown below:

C39ctf-for = 5'CATCCGGAGGACTTGCCTGCGTTGC
C39ctf-rev1 = 5'CTTGTCATCGTCATCCTTGTAATCGATATCATGATCTTTATAATCACCG CTCAGGTGCCGGGGTCCTC
C39ctf-rev2 = 5'TCTGTCGACTCTAGTCTAGACCGTGACCTCGGCTTTGCTGTTGGTGT GCTTGTCATCGTCATCCTTGTAATCG
C52ctf-for = 5'GCCTCGAGCGCGCCTGGTGTCTCAG
C52ctf-rev1 = 5'CTTGTCATCGTCATCCTTGTAATCGATATCATGATCTTTATAATCAC CAGTGCGCAGGTGCTGGGGCAG
C52ctf-rev2 = 5'TCTGTCGACTCTAGTTCAGACAGTCACTTCGTTCTTGCTGGCGTG CTTGTCATCGTCATCCTTGTAATCG
C59ctf-for = 5'CACTTCCTCGAGAACGGCCACGCAG
C59ctf-rev1 = 5'CTTGTCATCGTCATCCTTGTAATCGATATCATGATCTTTATAATCA CCTGTGTAGCAGGTGTGATGGC
C59ctf-rev2 = 5'TCTGTCGACTCTAGTTCACACGGTCACTTCAGTCTTGCTGGCGTG CTTGTCATCGTCATCCTTGTAATCG

## Western blot analysis and immunoprecipitation of proteins that were expressed in HEK293/T17 cells

For immunoprecipitation experiments, an STR-tested and authenticated HEK293/T17 cell line was used (American Type Culture Collection, CRL-11268, ATCC, USA). All cell cultures were tested for mycoplasma contamination prior to experiments using PCR Mycoplasma Test Kit I/C (PK-CA91-1024, PromoCell GmbH, Germany). Cell lines utilized in the study are not mentioned in the list of commonly misidentified cell lines maintained by the International Cell Line Authentication Committee. Both Western-blot and immunoprecipitation were performed as previously described (*Khodosevich et al., 2014*). Briefly, HEK293/T17 cells were co-transfected with CKAMP44, CKAMP39, CKAMP52, CKAMP59 (in pRK5) or EGFP expression (pEGFP-C1) plasmids together with a GluA1 or GluA2 expression plasmid (in pRK5). Two days post-transfection, protein was collected and affinity-purified with an agarose-bound flag antibody (Anti-FLAG M2 Affinity Gel, Sigma-Aldrich, Germany) as previously described (*Khodosevich et al., 2014*). For immunoprecipitation we used 350 µg of total protein. Immunoprecipitated proteins were eluted by 50 µl of 3Xflag peptide solution (Sigma-Aldrich, Germany). Denatured whole protein (6-10 µg) and immunoprecipitated (10-25 µl) samples were separated by SDS-PAGE and transferred onto PVDF membranes that were probed with the mouse anti-flag M2 antibody (1:2000, F1804, Sigma-Aldrich, Germany) and rabbit GluA1 (1:1000, Santa Cruz, Germany) or mouse GluA2 antibody (MAB397, 1:500, Millipore, Billerica, MA, USA)

## Cell surface biotinylation assay

For cell surface biotinylation assay, HEK293/T17 cells were co-transfected with pRK5-CKAMP39, -CKAMP52, -CKAMP59 or pAcGFP1-Mem (Clontech-Takara Bio, France) plasmids together with pRK5-GluA1-flip or pRK5-GluA2(Q)-flip plasmids using Lipofectamine 2000 reagent (Invitrogen, Germany). Forty-eight hours after transfection, cells were washed once with ice-cold PBS (pH 8.0). Cells were biotinylated at room temperature for 30 min with 0.8 mM solution of EZ-Link Sulfo-NHS-SS-Biotin reagent (Pierce, Rockford, IL) prepared in PBS (pH 8.0). Cells were subsequently washed with 50 mM Tris (pH 8.0) to quench any non-reacted biotinylation reagent, and twice with ice-cold PBS (pH 8.0) to remove excess biotinylation reagent. To capture biotinylated surface proteins, total protein was collected and affinity-purified with EZview Red Streptavidin Affinity Gel (Sigma-Aldrich, Germany) using manufacturer's protocol. Denatured total (5 µl) and biotinylated surface protein (10 µl) samples were separated by SDS-PAGE, and transferred onto PVDF membranes that were probed with mouse anti-GluA1 antibody (1:1500) or mouse anti-GluA2 antibody (1: 500) (MAB2263 and MAB397, respectively, both from Millipore, Billerica, MA, USA). For loading control, mouse anti-beta Actin antibody (1:4000, MA5-15739, Thermo Fisher Scientific, Rockford, USA) was used. Surface protein was normalized for equal protein concentration. Relative quantification of surface GluA1 or

GluA2 expression was carried out by densitometry of western blots using ImageJ software (http://imagej.nih.gov/ij).

## HEK cell transfection and electrophysiology

HEK293/T17 cell lines stably expressing GluA1-flip or GluA2(Q)-flip were grown and maintained using standard protocols. For electrophysiological recordings, cells were transfected using Lipofect-amine 2000 (Invitrogen, Germany) and pRK5-CKAMP39, -CKAMP52 or -CKAMP59 together with pEGFP-C1 (Clontech-Takara Bio, France) or pEGFP-C1 alone. Cells were recorded 24–72 h post-transfection. Fast application of glutamate onto lifted HEK293/T17 cells was performed as described (*Jonas and Sakmann, 1992*) using theta glass tubing mounted onto a piezo translator. AMPAR-mediated currents were evoked by a 1 ms glutamate pulse for analyzing amplitude and deactivation, by a 100 ms glutamate pulse for analyzing desensitization and steady-state current amplitude, and by two 1ms pulses with 10, 30, 100, 300, 1000 and 3000 inter-event interval for analyzing recovery from desensitization. Application pipettes were tested by perfusing solutions with different salt concentrations through the two barrels onto open patch pipettes and recording current changes with 1 and 100 ms transitions of the application pipette. Only application pipettes with 20–80% rise times below 100 µs and with a reasonable symmetrical on- and offset were used. However, the expected solution exchange time is considerably slower with the use of whole cells instead of outside-out patches (*Barberis et al., 2008*). The application solution contained (in mM): 135 NaCl, 10 HEPES, 5.4 KCl, 1.8 CaCl2, 1 MgCl2, 5 glucose (pH 7.2). Whole-HEK293/T17 cell recordings were performed at room-temperature using pipettes pulled from borosilicate glass capillaries with a resistance of 3–5 MΩ when filled with the following solution (in mM): 120 Cs-gluconate, 10 CsCl, 8 NaCl, 10 HEPES, 10 phosphocreatine-Na, 0.3 GTP, 2 MgATP, 0.2 EGTA (pH 7.3, adjusted with NaOH). Liquid junction potentials were not corrected. AMPAR-current deactivation and desensitization were fitted with two exponentials, and the weighted tau ($\tau_w$) was calculated as $\tau_w = (\tau_f \times a_f) + (\tau_s \times a_s)$, where $a_f$ and $a_s$ are the relative amplitudes of the fast ($\tau_f$) and slow ($\tau_s$) exponential components.

## In situ hybridization

The in situ hybridization was done as described before (*von Engelhardt et al., 2015*). Briefly, horizontal brain sections from adult C57Bl/6 mice were cut on the cryostat (Leica Microsystems, Germany) and hybridized with one of the following radiolabeled oligodeoxyribonucleotide probes:

Ckamp39ins1 = 5′TGAGAAGTTCTGTCAGTGTCCTGGTCACCGTGCGCCGAGC
Ckamp52ins1 = 5′AATGTCAGCCAGAGCCCTGTGGATGTTCATCTCTCGCGGA
Ckamp59ins1 = 5′GCGGCATAGCACGCCAGTCGAGGTTGGAGGGCTTCATGGTGTT

The oligodeoxyribonucleotide probes were 3′ end-labeled by terminal deoxynucleotidetransfer-ase and (a)-[33]P-dATP (Hartmann Analytic, Germany). Brain sections were then hybridized over night in, 4 x× SSC (Saline-sodium citrate buffer, 0.6 M NaCl, 0.06 M sodium citrate), 50% formamide, 10% dextrane and 1 pg/µl labeled oligodeoxyribonucleotide probes at 42○° C and subsequently washed at 55○°C for 30 min, dehydrated and exposed to Kodak R X-omat AR film for 1 week.

## Oocyte preparation, electrophysiology and Western blot analysis

Stage V–VI *Xenopus laevis* oocytes were prepared, injected with cRNA as previously described (*Priel et al., 2005*). Whole-cell two electrode voltage clamp (TEVC) recordings were used for estimation of current amplitude before proceeding to patch-clamp recordings and for determination of EC50 values for glutamate and CTZ. TEVC recordings were performed at 17°C, at holding potential of −70mV, using GeneClamp500 connected to digidata1322A and pCLAMP8.2 (Axon Instruments). Data was analyzed by pCLAMP8.2 and ORIGIN 8 (Origin Lab Corp.) for estimation of the respective EC50s. For outside-out macropatch recordings the vitelline membrane was removed using forceps. Recordings were performed at 17°C, at membrane potential of -−120mV, using Axopatch 200B amplifier connected to digidata1322A and pCLAMP8.2 (Axon Instruments, Foster City, CA). For rapid solution exchanges, a double-barrel glass (theta tube) mounted on a piezoelectric translator (Burleigh, Fishers, NY) was used as previously described (*Priel et al., 2005*). Patch electrodes were fabricated from borosilicateglass with a low resistance of 0.3–1 MΩ. Receptor deactivation and desensitization were measured by applying glutamate (10 mM) for 1 ms and 500 ms, respectively. Recovery from desensitization was estimated with the two-pulse protocol in which a constant 100 ms

application of glutamate (10 mM) was followed by a 100 ms test pulse applied at different time intervals. Western-blot analysis was done as previously described (*Priel et al., 2005*) on protein homogenates from 10 oocytes for each sample. Blots were probed with anti-Flag antibody (1:2000; monoclonal anti-FLAG M2, Sigma-Aldrich, Israel) and visualized using ChemiDoc XRS system (Bio-Rad Laboratories). GluA1-flip and GluA2(Q)-flip were injected at 1ng cRNA/oocyte and CKAMP39, CKAMP44 & CKAMP52 were injected at 1, 3 & 5 ng cRNA/oocyte, respectively. At these conditions, CKAMP39, CKAMP44 & CKAMP52 exhibited comparable protein expression levels (*Figure 3A*) without a significant influence on total current amplitude compared to oocytes expressing the AMPAR alone (*Supplementary file 1A*), thereby allowing better comparison between the CKAMPs in modulating AMPAR properties. Higher amounts of CKAMP cRNA injections caused a reduction in total current amplitude manifested by a reduction in total protein expression as revealed by Western-blot analysis with anti-GluA1 and anti-GluA2/3 antibodies, respectively (not shown).

## Statistics

Data are presented as mean ± standard error of the mean (SEM) and as median ± interquartile range (IQR). Statistical differences between groups were examined by ANOVA, followed by Bonferroni test when the values showed a normal distribution, or by Kruskall-Wallis One Way ANOVA, followed by Dunn's method for multiple comparisons for non-Gaussian distributed values. Normality of data distribution was tested by Kolmogorov-Smirnov test and equal variance by Bartlett's test. Statistical analysis was performed using ORIGIN 8 (Origin Lab Corp.) or the GraphPad Prism version 5.00, GraphPad Software, San Diego, CA, USA, www.graphpad.com. P values < 0.05 were considered statistically significant (* = p < 0.05, ** = p < 0.01, *** = p < 0.001, **** = p < 0.0001).

## Acknowledgements

Supported by the DFG, grant EN 948/2-1 (JvE) and the Israel Science Foundation, grant no. 919/11 (YSB). Anton Schulmann received support from the MD/PhD program of the University Heidelberg. YSB and YS thank Hadas Ashkenazi and Jonathan Zorea for help with oocyte preparation and TEVC recordings.

## Additional information

### Funding

| Funder | Grant reference number | Author |
| --- | --- | --- |
| Israel Science Foundation | 919/11 | Yael Stern-Bach |
| Deutsche Forschungsgemeinschaft | EN 948/2-1 | Jakob von Engelhardt |

The funders had no role in study design, data collection and interpretation, or the decision to submit the work for publication.

### Author contributions

Paul Farrow, Konstantin Khodosevich, Yechiam Sapir, Anton Schulmann, Muhammad Aslam, Jakob von Engelhardt, Conception and design, Acquisition of data, Analysis and interpretation of data, Drafting or revising the article; Yael Stern-Bach, Hannah Monyer, Conception and design, Analysis and interpretation of data, Drafting or revising the article

### Author ORCIDs

Jakob von Engelhardt   http://orcid.org/0000-0003-3861-3294

### Decision letter and Author response
Decision letter https://doi.org/10.7554/eLife.09693.014
Author response https://doi.org/10.7554/eLife.09693.015

## Additional files

**Supplementary files**
• Supplementary file 1.
DOI: https://doi.org/10.7554/eLife.09693.012

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
