## [Decision Letter]

Thank you for submitting your work entitled "CKAMPs: a novel family of putative AMPA receptor auxiliary subunits" for peer review at *eLife*. Your submission has been favorably evaluated by Gary Westbrook (Senior editor) and three reviewers, one of whom is a member of our Board of Reviewing Editors.

The reviewers have discussed the reviews with one another and the Reviewing editor has drafted this decision to help you prepare a revised submission.

Summary:

All of the reviewers thought that the study is potentially interesting for the broad readership of *eLife*, and that your work provides a comprehensive analysis of the CKAMP family of AMPA receptor auxiliary factors. However, some major concerns should be carefully addressed in the revised version of the manuscript. In particular the location of bands related to the molecular weights of the various CKAMPs, for example CKAMP52 was unclear after comparing Figures 2B and 3A and needs further clarification. The reviewers also asked for surface biotinylation studies to determine if CKAMPs decrease surface expression of AMPA receptors in HEK293 cells as predicted by the authors. You will see that Reviewer 3 had many concerns regarding Figures 3-5, which may require clarifications in text or perhaps additional experiments. With respect to the CTZ experiments in Figure 4, the authors should clarify the text and explanations related to this entire Figure and in particular how the CTZ dose-response curves were performed. The reviewers understand that this study will be of considerable interest and it is therefore important to ensure that this first description is of the highest quality.

The main criticisms are listed below.

*Reviewer 1:*

It would be good if the authors could sum up future interesting questions which could be addressed such as the possible impact of the various AMPAR interacting proteins on the number of spines, synapse density and possible influence on the morphology of target cells.

*Reviewer 2:*

Because all of the conclusions drawn from results in heterologous cells rely on relatively equal expression of proteins it is crucial for the authors to address these discrepancies. The following issues should be addressed:

1) Proposal that CKAMP members constitute a distinct protein family from that of Shisa members.

Figure 1: I assume that the sequences listed are from mouse but it is not stated in the legend. Reference to signal sequences and transmembrane domains should indicate that they are predictions or else experimental validation should be provided.

In the Results section, the authors state that, "While CKAMP39 had only one splice variant, both CKAMP52 and CKAMP59 had two splice-isoforms (Figure 2—figure supplement 1). In the subsequent experiments, we utilized the most abundant versions of CKAMPs, i.e. CKAMP52 and CKAMP59 lacking exon 4." This observation is very interesting, especially if the primary amino acid sequence is altered upon addition of exon 4? In Figure 1, where is the additional sequence located upon inclusion of exon 4? Does this disrupt any of the protein-protein interacting domains such as the AMPA-R interacting domain or the PDZ motif? Do Shisa members exhibit similar splice variants with or without the inclusion of a variable exon?

2) Distribution of CKAMP members in brain and AMPA-R interacting properties.

Figure 2B: How much input is in panel at left, 10% or 100%? This information should be in the figure legend. Where exactly is the flag-tag inserted? This information should be in Figure 1.

In the Discussion, the authors state that, "The reduction of AMPAR mediated current amplitudes by all three CKAMPs in HEK293 cells was unexpected, since CKAMP44 has the opposite effect on AMPARs of granule cells, possibly by promoting trafficking of AMPARs to the cell surface." Because of the opposing results in HEK293 cells and granule cells, surface biotinylation should be carried out to determine if CKAMPs decrease surface expression of AMPA receptors in HEK293 cells as predicted.

Also in the Discussion, the authors state that, "Indeed, the higher molecular weight – especially of CKAMP52 – in oocytes than in HEK293 cells (Figure 2B and 3A) is suggestive of such a difference." This statement is confusing because the lower band of CKAMP52 is similar to the single band of CKAMP44 in oocytes (Figure 3A) but both CKAMP52 bands are higher than CKAMP44 in HEK293 cells (Figure 2B). The authors state that the two bands reflect glycosylated versions of CKAMP59 and CKAMP52 in HEK293 cells (Figure 2B); therefore, it is unclear how the even greater molecular weight of CKAMP52 (almost at 100 kDa marker in Figure 3A) is achieved.

*Reviewer 3:*

1) Figure 2 panel A, Adult brain. What happens to the CKAMP39 signal in the olfactory bulb between horizontal and sagittal sections? It's there in the horizontal but disappears in the sagittal. Also in Figure 2, why was GluA2 co-IP not done?

2) In Figure 3, a blot including +CKAMP59 should be shown even if it has poor expression. Also, the glutamate concentration used should be stated in the legend or figure. In Figures 3 and 5 the deactivation time constants for GluA2 and GluA1 are unusually slow. The GluA1 looks to be about 2.5 ms (Figure 3B). Previous reports indicate that GluA1 deactivation is between 1 and 0.5 ms (Tomita et al., 2005 Figure 4D; Priel et al., 2005 Figure 2C; Bowie 2002 Figure 4, Partin et al., 1996 Figure 4) and GluA2 around 1 ms (C

arbone and Plested, Table 1. Why are deactivation time constants reported here 2 to 5 fold slower?

3) Figure 3 uses mean +/- SEM but in Figure 5 the authors use median plus interquartile range. Why? Mean +/- SEM is standard. Also in Figure 5, panel D is not a recovery from desensitization experiment. Rather than long pulses to fully desensitize the channels, the authors use 1 ms pulse to evoke 'deactivation' responses. This makes the experiment more of a paired pulse depression analogue than measuring recovery from desensitization (because the whole population doesn't actually desensitize). The authors should clearly state how long the pulses are (and not simply in the Methods) and unify their data presentation between Figure 3 and 5 (i.e. use either linear or log scale, IEI or time for panel D x axis).

4) Although the authors use the open tip current method for measuring exchange, that only tells you about the exchange at the open tip, not solution exchange around the whole cell. See Barberis et al., 2008 Supplementary Material page 1 and Supplemental Figure 1). Given this, the authors should either repeat the experiments in patches, simply drop the HEK cell data and use only the oocytes, or insert a disclaimer to the effect of 'we know these aren't actually 1 ms pulses but they are only used to compare differences between CKAMPs'.

5) Figure 4A. There is no time scale for the traces. Also, if the authors want to measure steady-state (SS) current, then the traces should be shown reaching SS. As I understand the experiment, they compare the SS current observed with simultaneous glu and CTZ application (Figure 4A, a traces) versus pre-exposure to CTZ before glutamate (Figure 4A, b traces) as a control. Several things will happen if one applies glu and CTZ simultaneously apparently without CTZ pre-equilibration (a traces). The glu-bound channels will activate and quickly begin to enter the desensitized state. While CTZ binds to available dimer interfaces. Considering that CTZ binds the dimer interface, as the channels move from apo to open to desensitized states (and the dimer interface rearranges), the CTZ affinities of the various conformations of dimers will be dynamically changing during the agonist application. Amidst this complicated swirl of conformational changes, the desensitization rates of these various receptor conformations are also different (Figure 3C), which would be expected to alter the final SS current. Moreover, the CTZ potency is apparently also different (Figure 4B) so when the authors apply 100 uM CTZ to GluA1 it apparently occupies only about 45% of the receptors whereas at GluA1+39 it would occupy 10% of the receptors. These experiments are done in oocytes which have slow solution exchange so this experiment is not so much applying Glu + CTZ as it is slowly ramping up the concentration of glu + CTZ from zero to something apparently subsaturating. This makes it difficult to draw any mechanistic inference. Confusion is increased by the use of GluA3 in this part of the paper. Overall, it is not clear how much Figure 4A adds to the manuscript as currently presented.

6) Figure 4B. Where is +CKAMP 44 or data for GluA2? Also, it is surprising that the EC50 for CTZ at GluA1 flip is 120 uM. Prior studies reported a value of 7 uM (Partin et al., 1994 Figure 2). Please address Figure 4 panels B and C (also in the text). In addition, the authors should use "potency" not "affinity". Affinity is only for kd. See Colquhoun (1998).

7) Figure 4D. Because CKAMP44 and 52 appear to 'compete' in this CTZ experiment, it does not provide evidence they 'form a protein family' (subsection “Novel CKAMP members modify AMPAR-mediated currents in heterologous”, third paragraph). TARPs or cornichon might do the same thing in such an assay. I suggest the authors remove this experiment. Should they wish to address this point, they could use TARPs or cornichon instead of Neto as a control and use GluA1 or 2 instead of 3.

In summary, Figure 4 could simply be CTZ dose-response curves (with explanation of how the curves were obtained, what conc. of glu for example) on GluA1, +39, +44 and +52 and GluA2 39, +44 and +52 as well as glutamate dose response curves with the same constructs.

---

## [Author Response]

We thank the reviewers for their constructive input and specific suggestions, which were instrumental in designing new experiments that helped improve the manuscript. We performed a number of additional experiments to address all concerns that were raised by the reviewers. One of the major criticisms raised

by Reviewer 2 was the lack of sufficient explanation for the reduction of AMPAR-mediated current amplitude by CKAMP family members. As suggested by the reviewer, we performed a biotinylation assay and found that CKAMP family members indeed reduce the number of surface AMPARs. Interestingly,

changes in total AMPAR expression and in the ratio of surface/total protein suggest that this reduction was in part due to an overall reduction in AMPAR protein, and in part due to a reduction in forward trafficking and/or stability of AMPARs on the cell surface. Reviewer 3 was in particular concerned with the interpretation of several experiments shown in Figure 4. To address his/her criticism, we performed a series of new experiments to estimate the influence of CKAMP family members on glutamate and cyclothiazide potency. In fact, nearly all panels of Figure 4 were replaced by panels showing new data providing evidence that all CKAMP family members increase glutamate potency and CKAMP39 and CKAMP52 decrease cyclothiazide potency.

We here provide a list of the figure panels presenting results that were not in the initial version:

Figure 2C: Co-IP of GluA2 with CKAMP family members

Figure 3A: New Western blot of CKAMP family members using the same blotting conditions and size marker that were also used for analyzing CKAMP family members expressed in HEK293 cells (i.e. Fig.2B and C).

Figure 4A: Investigation of glutamate potency (GluA1 and GluA2)

Figure 4B: Investigation of cyclothiazide potency (GluA1 and GluA2)

Figure 5—figure supplement 1: Biotinyltion assay for the quantification of surface and total GluA1 and GluA2 in HEK293 cells co-expressing CKAMP family members.

Reviewer 1:

*It would be good if the authors could sum up future interesting questions which could be addressed such as the possible impact of the various AMPAR interacting proteins on the number of spines, synapse density and possible influence on the morphology of target cells.*

We added future perspectives on the function of CKAMP family members in the Discussion.

Reviewer 2:

*1) Proposal that CKAMP members constitute a distinct protein family from that of Shisa members. Figure 1: I assume that the sequences listed are from mouse but it is not stated in the legend. Reference to signal sequences and transmembrane domains should indicate that they are predictions or else experimental validation should be provided.*

We added in the figure legends the information that the sequences refer to mouse proteins and that signal peptide and transmembrane domains are predicted.

*In the Results section, the authors state that, "While CKAMP39 had only one splice variant, both CKAMP52 and CKAMP59 had two splice-isoforms (Figure 2—figure supplement 1). In the subsequent experiments, we utilized the most abundant versions of CKAMPs, i.e. CKAMP52 and CKAMP59 lacking exon 4." This observation is very interesting, especially if the primary amino acid sequence is altered upon addition of exon 4? In Figure 1, where is the additional sequence located upon inclusion of exon 4? Does this disrupt any of the protein-protein interacting domains such as the AMPA-R interacting domain or the PDZ motif? Do Shisa members exhibit similar splice variants with or without the inclusion of a variable exon?*

We agree that it is interesting that there are two splice variants of CKAMP52 and CKAMP59. In fact, there are also two splice variants of CKAMP44. At the moment we did not study the possible function of the alternatively spliced exons that encode for 32 and 17 amino-acids, respectively. However, the position of these amino-acids downstream of the AMPAR-interaction domain and upstream of the PDZ domain binding motif suggest that the spliced version lacking the alternatively spliced exon is unlikely to exhibit altered interaction with AMPARs or PDZ domain-containing proteins. No alternatively spliced mRNA of Shisa genes were reported. We included all required information in the text and figure legend of the manuscript.

*2) Distribution of CKAMP members in brain and AMPA-R interacting properties.*

*Figure 2B: How much input is in panel at left, 10% or 100%? This information should be in the figure legend. Where exactly is the flag-tag inserted? This information should be in Figure 1.*

We added the information in the figure legend of Figure 2B and in Figure 1.

*In the Discussion, the authors state that, "The reduction of AMPAR mediated current amplitudes by all three CKAMPs in HEK293 cells was unexpected, since CKAMP44 has the opposite effect on AMPARs of granule cells, possibly by promoting trafficking of AMPARs to the cell surface." Because of the opposing results in HEK293 cells and granule cells, surface biotinylation should be carried out to determine if CKAMPs decrease surface expression of AMPA receptors in HEK293 cells as predicted.*

We thank the reviewer for suggesting we perform a biotinylation assay, since this experiment was indeed important to explain the reduction of AMPAR-mediated current amplitudes in HEK293 cells co-expressing CKAMP family members. Thus, the biotinylation assay revealed that CKAMP family members decrease the surface expression of AMPARs in HEK293 cells. Interestingly, this reduction could be explained in part by a reduction in total GluA1 and GluA2 expression and in part by a reduction of the forward trafficking or surface stability of AMPAPs, i.e. there was a reduction of total AMPAR expression and in addition a reduction in the ratio of surface/total AMPARs in cells expressing CKAMP family members. Finally, CKAMP family members seem to reduce the forward trafficking or surface stability in a subunit specific manner since the ratio of surface/total protein was reduced only for GluA2, but not for GluA1. In fact, CKAMP52 even increased the ratio of surface/total GluA1, thus appears to promote forward trafficking or surface stability of GluA1. We added a paragraph in which we describe the data and interpretation of these experiments.

*Also in the Discussion, the authors state that, "Indeed, the higher molecular weight – especially of CKAMP52 – in oocytes than in HEK293 cells (Figure 2B and 3A) is suggestive of such a difference." This statement is confusing because the lower band of CKAMP52 is similar to the single band of CKAMP44 in oocytes (Figure 3A) but both CKAMP52 bands are higher than CKAMP44 in HEK293 cells (Figure 2B). The authors state that the two bands reflect glycosylated versions of CKAMP59 and CKAMP52 in HEK293 cells (Figure 2B); therefore, it is unclear how the even greater molecular weight of CKAMP52 (almost at 100 kDa marker in Figure 3A) is achieved.*

To rule out that differences in the blotting conditions explain the higher molecular weight in oocytes than in HEK293 cells, we performed repeatedly Western blots of CKAMPs in oocytes exactly reproducing conditions (buffers, incubation time etc.) that were employed to detect CKAMPs in HEK293 cells. However, the molecular weight of CKAMPs expressed oocytes is still higher than that of CKAMPs expressed in HEK293 cells, and in both expression systems the molecular weight is bigger than the calculated weight of CKAMPs. Since the same CKAMP constructs were used in oocytes and HEK293 cells, the most parsimonious explanation for this finding is a difference in the post-translational modification in mammalian and amphibian cells. However, the reviewer is correct that the statement about potential differences in posttranslational modifications might be confusing. Since we cannot prove that posttranslational modifications are the cause for the small difference in the observed function of CKAMPs in oocytes and HEK293 cells, we deleted this sentence from the Discussion.

Reviewer 3:

*1) Figure 2 panel A, Adult brain. What happens to the CKAMP39 signal in the olfactory bulb between horizontal and sagittal sections? It's there in the horizontal but disappears in the sagittal.*

The discrepancy of the CKAMP39 signals on the horizontal and sagittal sections is due to the fact that the olfactory bulb was severed from the sagittally sliced brain. We now show a section in which the olfactory bulb is present.

*Also in Figure 2, why was GluA2 co-IP not done?*

We performed additional experiments with Co-IP of GluA2 (shown in Figure 2B).

*2) In Figure 3, a blot including +CKAMP59 should be shown even if it has poor expression.*

We have added the blot that shows the absence of CKAMP59 expression in oocytes.

*Also, the glutamate concentration used should be stated in the legend or figure.*

We added the missing information.

*In Figures 3 and 5 the deactivation time constants for GluA2 and GluA1 are unusually slow. The GluA1 looks to be about 2.5 ms (Figure 3B). Previous reports indicate that GluA1 deactivation is between 1 and 0.5 ms (Tomita et al., 2005 Figure 4D; Priel et al., 2005 Figure 2C; Bowie 2002 Figure 4, Partin et al., 1996 Figure 4) and GluA2 around 1 ms (Carbone and Plested, Table 1). Why are deactivation time constants reported here 2 to 5 fold slower?*

The main reason for slow deactivation time constants is that we used macropatches (oocyte experiments) and whole-cells (HEK293 experiments) for quantification of the influence of CKAMPs on AMPAR kinetics. As noted also by Reviewer 3 (comment 4), solution exchange is significantly slower when applying glutamate onto whole-cells or macropatches compared to the fast solution exchange time when using small outside-out patches. In addition, room temperature was relatively low (we use shared facilities that require a constant temperature of 17°C) when oocyte experiments were performed, which would also slow deactivation kinetics.

We could not use outside-out patches since AMPAR-mediated current amplitudes were reduced too strongly in patches of cells that co-expressed CKAMPs. A reliable quantification of AMPAR kinetics was not possible from patches with very small current amplitudes. Thus, these experiments are indeed suboptimal for quantifying the "real" influence of CKAMP on AMPAR deactivation kinetics. However, we believe that we still can infer from these experiments that CKAMP39 and CKAMP52 slow AMPAR deactivation.

*3) Figure 3 uses mean +/- SEM but in Figure 5 the authors use median plus interquartile range. Why? Mean +/- SEM is standard.*

Data from Figure 5 were not normally distributed. Thus, we tested for statistical differences between groups by Kruskall-Wallis One Way ANOVA. Accordingly, we decided to show median + IQR for these not normally distributed data. However, if the reviewer believes that we should be consistent in the way how to present data, we could change to mean +/- SD also in this figure.

*Also in Figure 5, panel D is not a recovery from desensitization experiment. Rather than long pulses to fully desensitize the channels, the authors use 1 ms pulse to evoke 'deactivation' responses. This makes the experiment more of a paired pulse depression analogue than measuring recovery from desensitization (because the whole population doesn't actually desensitize). The authors should clearly state how long the pulses are (and not simply in the Methods) and unify their data presentation between Figure 3 and 5 (i.e. use either linear or log scale, IEI or time for panel D x axis).*

We did not perform a classical recovery from desensitization experiment but wanted to get an idea about the recovery from desensitization of AMPA receptors that are exposed to glutamate for a short time (similar to the short presence of glutamate in the synaptic cleft). From our previous studies we knew that CKAMP44 reduces synaptic short-term plasticity in synapses of dentate gyrus granule cells despite the short presence of glutamate in the synaptic cleft. CKAMP39 in fact slows recovery from desensitization similar to CKAMP44 (Khodosevich et al., Neuron 2014) suggesting that it also could influence short-term plasticity in synapses with high release probability. We have added more information to the Results about the experiment and unified data presentation of Figures 3 and 5.

*4) Although the authors use the open tip current method for measuring exchange, that only tells you about the exchange at the open tip, not solution exchange around the whole cell. See Barberis et al., 2008 Supplementary Material page 1 and Supplemental Figure 1). Given this, the authors should either repeat the experiments in patches, simply drop the HEK cell data and use only the oocytes, or insert a disclaimer to the effect of 'we know these aren't actually 1 ms pulses but they are only used to compare differences between CKAMPs'.*

Indeed, an analysis of AMPA receptors with patches would have been better. Unfortunately, we could not perform the analysis on patches, because of the dramatic reduction in AMPA receptor number on the cell surface in cells that co-expressed CKAMPs. In many patches, currents were simply too small for an analysis of kinetics. However, we would like to show the data since we believe that despite the comparably slow application the results of the HEK293 cell experiments add some information about differences in the function of CKAMPs. Thus, we would prefer to add – as suggested – the information that glutamate pulses are slower when applied onto whole-cells instead of patches.

*5) Figure 4A. There is no time scale for the traces. Also, if the authors want to measure steady-state (SS) current, then the traces should be shown reaching SS. As I understand the experiment, they compare the SS current observed with simultaneous glu and CTZ application (Figure 4A, a traces) versus pre-exposure to CTZ before glutamate (Figure 4A, b traces) as a control. Several things will happen if one applies glu and CTZ simultaneously apparently without CTZ pre-equilibration (a traces). The glu-bound channels will activate and quickly begin to enter the desensitized state. While CTZ binds to available dimer interfaces. Considering that CTZ binds the dimer interface, as the channels move from apo to open to desensitized states (and the dimer interface rearranges), the CTZ affinities of the various conformations of dimers will be dynamically changing during the agonist application. Amidst this complicated swirl of conformational changes, the desensitization rates of these various receptor conformations are also different (Figure 3C), which would be expected to alter the final SS current. Moreover, the CTZ potency is apparently also different (Figure 4B) so when the authors apply 100 uM CTZ to GluA1 it apparently occupies only about 45% of the receptors whereas at GluA1+39 it would occupy 10% of the receptors. These experiments are done in oocytes which have slow solution exchange so this experiment is not so much applying Glu + CTZ as it is slowly ramping up the concentration of glu + CTZ from zero to something apparently subsaturating. This makes it difficult to draw any mechanistic inference. Confusion is increased by the use of GluA3 in this part of the paper. Overall, it is not clear how much Figure 4A adds to the manuscript as currently presented.*

We agree with the reviewer. It is rather difficult to draw conclusions from the experiment with co-application of glutamate and CTZ. Thus, we performed additional experiments in which we quantified the CTZ EC50 for GluA1 and GluA2 by applying glutamate and the respective CTZ concentration onto oocytes after 10 sec application of CTZ alone.

*6) Figure 4B. Where is +CKAMP 44 or data for GluA2? Also, it is surprising that the EC50 for CTZ at GluA1 flip is 120 uM. Prior studies reported a value of 7 uM (Partin et al., 1994 Figure 2). Please address Figure 4 panels B and C (also in the text). In addition, the authors should use "potency" not "affinity". Affinity is only for kd. See Colquhoun (1998).*

The lower CTZ EC50 is explained by the co-application protocol. As noted by the reviewer in the previous comment, current amplitude was quantified 10 sec after glutamate + CTZ co-application; indeed no real SS current was reached. We repeated these experiments by including 10 sec pre-incubation of the oocyte with CTZ alone, before its co-application with glutamate to reach an equilibrium SS current (as done by Partin et al., 1994). Under these conditions a significant lower EC50 was found. The difference of the CTZ EC50 in our study (15 µM) and the study of Partin et al. (7 µM) might be due to technical differences (e.g. we used 1 mM glutamate and Partin et al. used 300 µM kainate to activate AMPARs). We corrected for potency instead of affinity.

*7) Figure 4D. Because CKAMP44 and 52 appear to 'compete' in this CTZ experiment, it does not provide evidence they 'form a protein family' (subsection “Novel CKAMP members modify AMPAR-mediated currents in heterologous”, third paragraph). TARPs or cornichon might do the same thing in such an assay. I suggest the authors remove this experiment. Should they wish to address this point, they could use TARPs or cornichon instead of Neto as a control and use GluA1 or 2 instead of 3.*

We agree with the reviewer and deleted the experiment.

*In summary, Figure 4 could simply be CTZ dose-response curves (with explanation of how the curves were obtained, what conc. of glu for example) on GluA1, +39, +44 and +52 and GluA2 39, +44 and +52 as well as glutamate dose response curves with the same constructs.*

We followed the suggestion of the reviewer.